# Characteristics and Outcomes of Hospitalized Patients with Histoplasmosis: Comparison of Immunocompromised and Non-Immunocompromised Adult Patients

**DOI:** 10.3390/jof11090671

**Published:** 2025-09-12

**Authors:** Liam M. Dalton, Carol A. Kauffman, Marisa H. Miceli

**Affiliations:** Division of Infectious Diseases, Department of Internal Medicine, University of Michigan Medical Center, Ann Arbor, MI 48109, USA

**Keywords:** histoplasmosis, immunocompromised host, disseminated infection, pulmonary infection

## Abstract

We sought to investigate the role of immunocompromise in patients with newly diagnosed histoplasmosis in an era when AIDS is less prevalent. We performed a retrospective comparison of immunocompromised and non-immunocompromised adults hospitalized at Michigan Medicine from 2015 to 2024. Of 51 patients, 37 (73%) were immunocompromised, 32 from solid organ transplantation or tumor necrosis factor antagonist/disease-modifying anti-rheumatic drugs. Of these 37, 34 had disseminated and 3 had pulmonary histoplasmosis; of the 14 non-immunocompromised patients, 8 had disseminated and 6 had pulmonary histoplasmosis, *p* = 0.004. Fever was the only symptom/sign that was more common in the immunocompromised cohort (86% vs. 36%, *p* = 0.003). Laboratory/radiological studies showed no major differences between immunocompromised and non-immunocompromised cohorts. *Histoplasma* urinary antigen was positive for all immunocompromised vs. 79% non-immunocompromised patients, *p* = 0.003. Median antigen levels were 17.5 (IQR 6.2–19.7) ng/mL for immunocompromised vs. 1.9 (0.6–19.7) ng/mL for non-immunocompromised patients, *p* = 0.004. Cultures for *Histoplasma* were more often positive in the immunocompromised cohort, *p* = 0.025. All-cause 90-day mortality was 14% in each cohort (five immunocompromised and two non-immunocompromised patients); all deaths occurred in those with disseminated histoplasmosis, and four were in the first week of hospitalization. Disseminated histoplasmosis in both immunocompromised and non-immunocompromised patients continues to be a serious, often fatal infection.

## 1. Introduction

The manifestations of infection with *Histoplasma capsulatum* vary widely, from mild respiratory infection to severe pulmonary or disseminated illness, depending on the balance between exposure to the organism in the environment and the immune status of the host [1]. Early studies called attention to the importance of the immune response in the 1970s, typified by the reticuloendothelial system in response to exposure to *H. capsulatum*, and more recent studies have elucidated the complex mechanisms involved in both the innate and adaptive immune responses to *H. capsulatum* [2]. With the emergence of HIV, new perspectives were gained on the importance of CD4 lymphocytes in the immune response to this organism, and most reports of histoplasmosis in immunosuppressed patients have focused on persons with HIV [3,4,5,6]. An increased risk of histoplasmosis has also been reported in patients who are immunosuppressed after receiving a solid organ transplant and in patients who have been prescribed an agent to inhibit the activity of tumor necrosis factor (TNF), which is essential in the response to infection with *H. capsulatum* [7,8,9,10,11,12,13]. Separate from reports dealing with these specific patient populations, few recent studies have looked at a broader patient population with respect to the effects of immune compromise in response to histoplasmosis [14,15,16,17]. We undertook a contemporaneous analysis of the role of immune compromise in patients who required hospitalization for newly diagnosed histoplasmosis in our medical center. We noted possible environmental exposures, clinical presentations, radiological and diagnostic laboratory tests, treatment, and outcomes in adults who were immunocompromised when compared with those who were presumed to be immunocompetent.

## 2. Materials and Methods

### 2.1. Patients and Setting

This retrospective study was conducted at Michigan Medicine, a 1000-bed tertiary care hospital in southeastern Michigan. All patients 18 years of age and older who were hospitalized between 1/1/15 and 2/1/24 were eligible for inclusion in the study if they had a new diagnosis of histoplasmosis defined by a positive culture for *H. capsulatum*, histopathology showing 2–4 µm budding yeasts, or a positive enzyme immunoassay (EIA) for *Histoplasma* antigen in urine or serum.

Patients were stratified into immunocompromised and non-immunocompromised cohorts. Immunocompromise was defined as solid organ or hematopoietic cell transplant, hematological or solid tumor malignancy treated with chemotherapy in the preceding 90 days, HIV infection with CD4 count <200 cells/µL, rheumatologic, gastrointestinal, or cutaneous diseases treated with immunosuppressive disease modifying anti-rheumatic drugs in the 90 days prior to diagnosis, or high-dose corticosteroid therapy defined as the equivalent of ≥0.3 mg/kg/day of corticosteroids for ≥3 weeks in the 60 days prior to the diagnosis of histoplasmosis.

Patients were classified as having proven or probable histoplasmosis according to EORTC/MSGERC definitions [18]. Patients were noted to have only localized pulmonary infection when no evidence for histoplasmosis was found outside of the lungs or mediastinum. Disseminated histoplasmosis was defined as infection involving organs in non-contiguous sites in addition to or other than the lungs or mediastinum.

Data Direct, an in-house electronic medical record data repository, was utilized to identify inpatients with histoplasmosis [19]. All data were stored in a secure REDCap (Research Electronic Data Capture) database for further analysis [20]. Data collected from the electronic medical record, including patient demographics, types of immunocompromise, risk factors for histoplasmosis, environmental exposure, presenting symptoms and signs, diagnostic studies, treatment approaches, and outcomes, were compared between immunocompromised and non-immunocompromised cohorts. This study was approved by the University of Michigan Institutional Review Board; informed consent was deferred due to the retrospective nature of this study.

### 2.2. Statistical Analysis

Results are presented as median (interquartile range [IQR]). The Wilcoxon Rank-Sum test was used to determine differences between groups with binomial measures, and the Chi-square test was used for categorical variables. Statistical significance was defined as *p* < 0.05. Analyses were completed using R, version 4.3.2 (R Foundation for Statistical Computing, Vienna, Austria).

## 3. Results

### 3.1. Patients

Fifty-one hospitalized patients were found to have newly diagnosed histoplasmosis. Among these 51 patients, 37 (73%) were immunocompromised, and 14 (27%) were not immunocompromised. The median age for those who were immunocompromised was 45 (31.5–60.5) years compared with 61 (41–63) years for those who were not immunocompromised, *p* = 0.12, and 63% of patients were men (Table 1). The most common immunocompromising condition was receipt of a solid organ transplant, noted in 19 (37%) patients. A total of 17 (33%) patients had a rheumatologic or gastrointestinal disease, for which 13 were on a variety of disease-modifying anti-rheumatic drugs. Only one patient appeared to have no underlying illness. Of the 37 patients who were immunocompromised, 34 (92%) had disseminated histoplasmosis and 3 (8%) had only pulmonary infection, while 8 (57%) of the 14 patients who were not immunocompromised had disseminated disease, and 6 (43%) had only pulmonary infection, *p* = 0.004.

All non-immunocompromised patients with disseminated histoplasmosis had chronic illnesses, including solid tumors, diabetes, sarcoidosis, and rheumatoid arthritis. None of the patients with solid tumors, rheumatoid arthritis, or sarcoidosis were on immunosuppressive medications, as defined in the Methods, and all of the patients with diabetes had well-controlled disease.

Of the 19 patients who had received a solid organ transplant, 18 had disseminated histoplasmosis, and 1 patient had only pulmonary histoplasmosis. The median time from transplant to hospitalization for histoplasmosis was 1962 (955–2643) days. Two patients developed histoplasmosis within 90 days of receiving their transplant; induction therapy in those two patients was with anti-thymocyte globulin in one and methylprednisolone in the other. In no patient did histoplasmosis lead to graft dysfunction or loss.

### 3.2. Epidemiology and Exposure History

Dates of hospitalization for the immunocompromised cohort clustered in the late fall and early winter months, but for non-immunocompromised patients were more evenly spread throughout the year, *p* = 0.02 (Figure 1). The geographical distribution of patients’ residences was primarily concentrated in the catchment area of the medical center (Figure 2). Possible exposures to *H. capsulatum* were elicited in 15 (41%) of the immunocompromised cohort and 6 (43%) of the non-immunocompromised cohort. These included living in a rural area, often in an older farmhouse, tree trimming, landscaping, working in a factory in which wood was chipped or processed, raising chickens, and visiting or living in rural Mexico prior to the onset of illness.

### 3.3. Clinical Manifestations

Fever and fatigue were the most common symptoms noted by both immunocompromised and non-immunocompromised persons on admission to the hospital. Fever was documented in 32 (86%) of patients who were immunocompromised, but in only 5 (36%) of those who were not, *p* = 0.003; fatigue was present in 23 (62%) of the immunocompromised cohort and 12 (86%) of the non-immunocompromised cohort, *p* = 0.11 (Table 2). Respiratory symptoms did not differ significantly between the immunocompromised and the non-immunocompromised. Four of the nine (44%) patients with pulmonary histoplasmosis and 12 of the 42 (29%) patients with disseminated infection had cough and/or dyspnea, *p* = 0.35. Altered mental status was noted in seven patients, but documented central nervous system (CNS) involvement was noted in only one patient in each cohort.

### 3.4. Laboratory and Radiological Studies

Laboratory studies did not differ significantly between the immunocompromised and non-immunocompromised cohorts (Table 3). However, when comparing the nine patients who had only pulmonary involvement with the 42 patients who had disseminated histoplasmosis, several differences in laboratory tests were seen. Unexplained pancytopenia was not seen in patients with pulmonary infection, while 13 (31%) of those with disseminated histoplasmosis had pancytopenia, *p* = 0.05. One patient with pancytopenia had hemophagocytic lymphohistiocytosis (HLH). Liver enzymes more than two times the normal values that could not be explained by underlying illness were found in 21 (50%) patients with disseminated infection and in no patients with pulmonary histoplasmosis, *p* = 0.005.

Computed tomography (CT) of the thorax was available for all 51 patients (Table 4). The most common finding in both cohorts was pulmonary nodules, described in 20 (54%) of those who were immunocompromised and 7 (50%) of those who were not. Cavities were present in four (29%) non-immunocompromised patients, all of whom had only pulmonary infection, but only two (5%) of patients who were immunocompromised, both of whom had disseminated histoplasmosis, *p* = 0.04. In four patients, the cavities were small, single, and approximately 1 cm in diameter, but one patient in each cohort had larger cavities of 5–11 cm. For the two patients who had documented CNS disease, magnetic resonance imaging (MRI) showed meningeal thickening and multiple enhancing cerebral lesions.

### 3.5. Histoplasma Diagnostic Studies

*Histoplasma* urinary antigen testing by EIA was obtained in all patients and was positive in all 37 immunocompromised patients and in 11 (79%) of the 14 non-immunocompromised patients, *p* = 0.003 (Table 5). Although ordered less often, *Histoplasma* serum antigen was positive in all 22 immunocompromised patients who were tested, but only 66% of the nine non-immunocompromised patients who were tested had a positive test, *p* = 0.004. CSF from 1 of 10 patients tested positive for *Histoplasma* antigen.

Significant differences between the two cohorts were found in both urine and serum antigen values. Among immunocompromised patients, the median urine antigen value was 17.5 (6.2–19.7) ng/mL compared with 1.9 (0.6–19.7) ng/mL for non-immunocompromised patients, *p* = 0.004. Median serum antigen value was 19 (14.5–20) ng/mL for those who were immunocompromised versus 1.8 (0–18.9) ng/mL for those who were non-immunocompromised, *p* = 0.002. Patients with disseminated infection had significantly higher urine antigen values, 17.4 (6.2–19) ng/mL, than those with only pulmonary infection, 2.7 (0.3–13.7) ng/mL, *p* = 0.02. Similarly, serum antigen values were 19 (9.9–19.7) ng/mL for those with disseminated infection and 1.3 (0–19) ng/mL for those with only pulmonary infection, *p* = 0.03

All but four patients, three of whom were immunocompromised, had cultures for fungi obtained from one or more sites (Table 5). The yield of isolating *H. capsulatum* from blood was 14/27 (52%) in the immunocompromised cohort and 3/10 (30%) in the non-immunocompromised cohort, and occurred only in those with disseminated disease. Nine of eleven (82%) cultures of bronchoalveolar lavage (BAL) fluid from the immunocompromised cohort yielded *H. capsulatum*, as did the one BAL sample from a patient who was not immunocompromised. CSF cultures from one of ten patients yielded *H. capsulatum*. Overall, the yield of positive cultures was greater in the immunocompromised cohort, *p* = 0.025.

Antibodies against *H. capsulatum* tested by complement fixation and immunodiffusion did not differ significantly between the two cohorts. Positive results were documented after the diagnosis had been established by other means in most patients.

### 3.6. Treatment and Outcomes

Liposomal amphotericin B (L-AmB) was given as initial therapy for 45 of the 51 (88%) patients, including 34 (92%) of the immunocompromised cohort and 11 (79%) of the non-immunocompromised cohort. Five patients, including two in the immunocompromised cohort (one each with pulmonary or disseminated disease) and three in the non-immunocompromised cohort (one with pulmonary and two with disseminated disease), were treated only with itraconazole. One immunocompromised patient, who was air-lifted to our medical center, died on admission before treatment could be given.

Fifteen (41%) patients in the immunocompromised cohort and five (36%) in the non-immunocompromised cohort were admitted to the intensive care unit (ICU), *p* = 0.33. Only nine patients required mechanical ventilation, five had a SOFA score of 2 or greater, and one patient was diagnosed with acute respiratory distress syndrome (ARDS). Four (44%) patients who had pulmonary histoplasmosis were admitted to the ICU compared with sixteen (38%) who had disseminated infection, *p* = 0.12. Median length of hospital stay was 13 (7–25) days for the immunocompromised cohort and 12 (9–20) days for the non-immunocompromised cohort, *p* = 0.65.

All-cause 90-day mortality was 14%; all seven patients who died had disseminated infection,5 were immunocompromised and two were non-immunocompromised. The one patient who had HLH died; the two patients with CNS involvement with histoplasmosis responded well to antifungal therapy. Four patients, three of whom were immunocompromised, died within the first week of hospitalization.

## 4. Discussion

Over several decades, we have noticed changes in the demographics of patients who were hospitalized for histoplasmosis at our medical center. Not only did more patients appear to be immunocompromised, but the underlying reasons for immunocompromise had changed. We sought to better define for the decade since 2015 whether the contemporaneous population of immunocompromised patients manifested differences in presentation and outcomes when compared with patients who were not immunocompromised. In comparison with prior studies that focused on specific at-risk populations [3,4,5,6,7,8,9,10,11,12,13], we included all patients who were hospitalized with histoplasmosis at our institution. Our data complement several earlier studies that analyzed all cases of histoplasmosis at a specific medical center but differ in that our study included only those patients who required hospitalization, whereas others included both inpatients and outpatients [14,15].

In comparison with several other studies, we noted a few patients with HIV as a cause of immune compromise [15,16]. Although listed as an AIDS-defining illness, currently in the United States, HIV is noted less often as a risk factor for histoplasmosis. In keeping with this change in demographics, we did not see several complications, such as sepsis, disseminated intravascular coagulation, and multiple cutaneous lesions that often have been described in patients with advanced HIV [3,21,22,23]. Almost all of the patients with these findings had extremely low CD4 counts, likely to account for the severe manifestations described.

Immunocompromise in our patients was primarily associated with receipt of a solid organ transplant or treatment with TNF antagonists and other disease-modifying anti-rheumatic drugs for rheumatologic or gastrointestinal diseases, reflecting the usual spectrum of immunocompromised patients seen in many hospitals today. All but three patients in the immunocompromised cohort had disseminated histoplasmosis, which is similar to other reports [14,15,16,24]. However, disseminated infection was also noted in more than half of the patients who did not meet the criteria for immunocompromise.

Of note, no patients presented with the classical disease manifestations of chronic cavitary pulmonary histoplasmosis, usually seen in persons with emphysema, or progressive disseminated histoplasmosis, as described in middle-aged to elderly persons with no known immunocompromise [25,26,27,28]. Our impression is that we now see fewer patients with these forms of histoplasmosis. We suspect that increased recognition and treatment earlier in *Histoplasma* infection, aided greatly by the widespread availability of *Histoplasma* antigen assays, has contributed to this decrease in these chronic forms of histoplasmosis.

In this study, we saw very few patients who developed several serious complications of histoplasmosis. There were no patients with endocarditis, a rare complication seen most often in older men with prosthetic cardiac valves [29], and we cared for only one patient who developed HLH related to histoplasmosis [30]. There were two patients who had CNS involvement as one manifestation of disseminated disease, a complication noted more often in patients with HIV [31].

Fever was the only symptom that was significantly more common in those who were immunocompromised, which is similar to the observations of Franklin et al. [15]. Respiratory symptoms were seen in similar proportions in both cohorts and were as common in patients who had disseminated infection as in those who had only pulmonary involvement. Laboratory studies were similar in the two cohorts, but for patients who present with non-specific symptoms of fever and fatigue, the presence of pancytopenia and liver enzyme elevations should be viewed as clues to a diagnosis of possible disseminated histoplasmosis. The presence of cavities on CT scan was more common in the non-immunocompromised cohort, but no other radiological parameters differed significantly between the two cohorts. We suspect that this difference likely relates to non-immunocompromised patients having preservation of immune-driven necrosis of infected lung tissue, a process that is markedly reduced among immunocompromised patients.

We found that most admissions among immunocompromised persons were in the late fall and early winter, differing from admissions of persons who were non-immunocompromised and spread throughout the year. Other similar studies from single institutions have not detailed the time of occurrence of disease symptoms or hospitalizations [14,15,16,17,24]. Based on voluntary reporting in 2019 from health departments in only 13 states, the Centers for Disease Control and Prevention noted that the frequency of reported cases of histoplasmosis was relatively consistent throughout the year [32]. Because we only captured data on the date of hospital admission, rather than the onset of disease, which could have preceded admission by weeks, we cannot establish a relationship between the onset of disease and the seasonal trends that we found.

EIA testing for the galactomannan found in the *Histoplasma* cell wall has revolutionized the diagnosis of histoplasmosis, providing crucial diagnostic information before culture results are available [33]. The *Histoplasma* antigen test was uniformly positive in both urine and serum from all immunocompromised patients who were tested, and both sites were significantly more often positive in this cohort than in the non-immunocompromised cohort. Antigen values were notably higher in immunocompromised individuals, likely reflecting a greater fungal burden [15]. Very high urine antigen values above the limit of quantification were observed more often in patients who had disseminated infection. However, it is noteworthy that several patients with isolated pulmonary histoplasmosis also demonstrated antigen levels above the level of quantification, emphasizing that the magnitude of antigenemia or antigenuria cannot reliably distinguish between pulmonary and disseminated disease and that results must be interpreted within the broader clinical context.

Culture and histopathology remain the gold standards for proving the diagnosis of histoplasmosis [18]. However, histopathology is not always readily attainable, and growth of *Histoplasma* from tissue or body fluids can take up to six weeks. Despite these limitations, cultures, especially those obtained from blood samples that generally yield growth more quickly than other samples, are particularly useful for confirming disseminated disease in immunocompromised patients.

Mortality was limited to patients with disseminated infection, and in the immunocompromised cohort, more than half the deaths occurred within the first week of admission, emphasizing the importance of early diagnosis and prompt initiation of appropriate antifungal therapy. Delays in diagnosis of histoplasmosis can be attributed to several causes [34]. Access to rapid testing and reporting of *Histoplasma* antigen in urine and serum helps to obviate some of the delay [35]. More important, however, is early recognition of the symptoms, signs, and abnormalities in routinely ordered laboratory studies that suggest the possibility of disseminated histoplasmosis. A positive antigen test can confirm probable histoplasmosis and allow prompt initiation of antifungal therapy.

Our study has several important limitations. The sample size is relatively small, reflecting the patient population at only one medical center, and reliance on retrospective chart reviews could have introduced potential bias. We do not have data on the time of first symptoms before hospitalization nor on the course of treatment for histoplasmosis, which could have enhanced the value of the study.

In conclusion, we found that most patients who were hospitalized with histoplasmosis in the past decade and who were immunocompromised had received a solid organ transplant or were being treated with TNF antagonists or other disease-modifying anti-rheumatic drugs. Compared with non-immunocompromised patients, immunocompromised patients were significantly more likely to present with fever, experience disseminated histoplasmosis, have positive tests for *Histoplasma* antigen in both urine and serum, as well as higher median antigen values from both sites, and to have *H. capsulatum* recovered from cultures. However, there were no significant differences between immunocompromised and non-immunocompromised patients in regard to ICU admission rates, lengths of hospital stay, and all-cause 90-day mortality rates.

## Figures and Tables

**Figure 1 jof-11-00671-f001:**
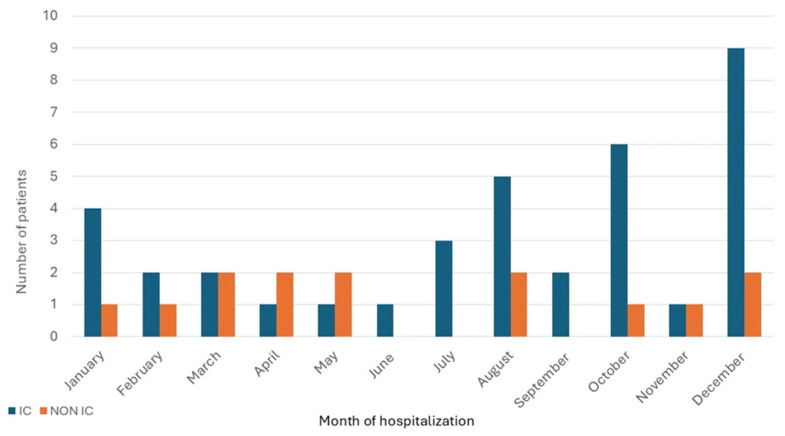
Distribution of patients with newly diagnosed histoplasmosis by month of hospitalization.

**Figure 2 jof-11-00671-f002:**
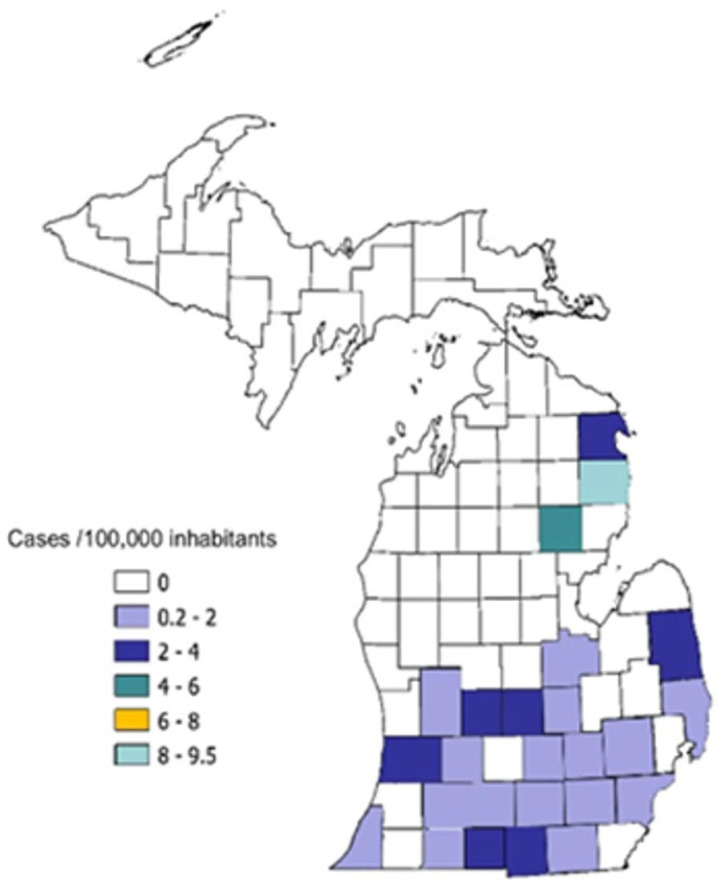
Geographical distribution of hospitalized patients in Michigan newly diagnosed with histoplasmosis by county of residence. Incidence rates are expressed as cases per 100,000 inhabitants, calculated using patient’s ZIP code at admission.

**Table 1 jof-11-00671-t001:** Demographics and underlying conditions in 51 hospitalized patients with a new diagnosis of histoplasmosis.

Characteristic	Immunocompromised n = 37	Non-Immunocompromised n = 14
Age: median (IQR)	45 (31.5–60.5)	61 (41–63)
Male	24	8
Underlying illness		
Diabetes mellitus	6	2
Chronic lung disease	5	2
HIV		
<200 CD4	2	0
>200 CD4	0	1
Solid organ transplant ^1^	19	0
Rheumatologic disease ^2^	9	3
Ulcerative colitis/Crohn’s disease	4	1
Sarcoidosis	1	2
Solid organ cancer	0	3
Leukemia/lymphoma	2	0
Immunosuppressive drugs		
Mycophenolate mofetil	19	-
Calcineurin inhibitor	17	-
Other disease-modifying anti-rheumatics drugs ^3^	10	-
Tumor necrosis factor antagonist	9	-
Corticosteroids ^4^	2	-
Histoplasmosis syndrome		
Pulmonary	3	6
Disseminated	34	8

IQR = Interquartile range. ^1^ Kidney: 14; liver: 3; lung: 2; heart: 2 (one patient each received both a liver and a kidney transplant and a kidney and a heart transplant). ^2^ Rheumatoid arthritis, dermatomyositis, systemic lupus erythematosus, mixed connective tissue disease, systemic sclerosis, ankylosing spondylitis. ^3^ Methotrexate, azathioprine, 6-mercaptopurine. ^4^ Equivalent of ≥0.3 mg/kg corticosteroids for ≥3 weeks in the prior 60 days.

**Table 2 jof-11-00671-t002:** Clinical manifestations in 51 patients hospitalized with a new diagnosis of histoplasmosis.

Symptoms/Signs	Immunocompromised (n = 37)	Nonimmunocompromised (n = 14)	*p* Value
Fever	32 (86%)	5 (36%)	0.003
Chills	14 (38%)	1 (7%)	0.07
Night sweats	7 (19%)	1 (7%)	0.66
Fatigue	23 (62%)	12(86%)	0.11
Weight loss	7 (19%)	5 (36%)	0.25
Shortness of breath	14 (38%)	3 (21%)	0.26
Cough	13 (35%)	5 (36%)	1
Pleuritic chest pain	0	2 (14%)	0.06
Abdominal pain	7 (19%)	3 (21%)	0.7
Skin lesions	0	0	-
Oral mucosal lesions	0	0	-
Headache	5 (14%)	0	0.3
Altered mental status	5 (14%)	2 (14%)	1

**Table 3 jof-11-00671-t003:** Laboratory findings in 51 patients hospitalized with a new diagnosis of histoplasmosis.

Blood Test ^1^	Immunocompromised (n = 37)	Nonimmunocompromised (n = 14)	*p* Value
Hemoglobin (g/dL)	10.8 (9.1–12)	10.1 (8.8–11.2)	0.53
White blood cell (K/μL)	3.9 (2.6–6.1)	6.1 (3.6–12.4)	0.05
Platelets (K/μL)	132 (84–176)	207 (123–326)	0.06
Aspartate aminotransferase (U/L)	71 (34–189)	37 (22–78)	0.1
Alanine aminotransaminase (U/L)	64 (33–148)	44.5 (16–65)	0.1
Alkaline phosphatase (U/L)	118 (77–201)	141 (82–210)	0.89

^1^ Values are presented as median and interquartile range (IQR).

**Table 4 jof-11-00671-t004:** Findings on computed tomography (CT) of thorax in 51 patients hospitalized with a new diagnosis of histoplasmosis.

Abnormalities in CT Thorax	Immunocompromised (n = 37)	Nonimmunocompromised (n = 14)	*p* Value
Nodules	20 (54%)	7 (50%)	1
Reticulonodular infiltrates	10 (27%)	2 (14%)	0.47
Cavities	2 (5%)	4 (29%)	0.04
Ground glass opacities	7 (19%)	3 (21%)	1
Consolidation	9 (24%)	3 (21%)	1
Tree-in-bud opacities	2 (5%)	0	1
Mediastinal/hilar adenopathy	14 (38%)	2 (14%)	0.18
Mediastinal/hilar calcification	1 (3%)	1 (7%)	0.48

**Table 5 jof-11-00671-t005:** Diagnostic studies supporting the diagnosis of histoplasmosis in 51 patients hospitalized with a new diagnosis of histoplasmosis.

Diagnostic Study	Immunocompromised (n = 37) (+) Tests/pt. Tested	Nonimmunocompromised (n = 14) (+) Tests/pt. Tested	*p* Value
Culture	27/34 (79%)	6/13 (46%)	0.025
Blood	14/27	3/10	
Bronchoalveolar lavage	9/11	1/1	
Sputum	2/3	1/3	
Cerebrospinal fluid	0/7	1/3	
Other	2/4 ^1^	0/2 ^2^	
Histopathology	9/15 (60%)	3/10 (30%)	0.15
Serology ^3^	18/31 (58%)	8/12 (66%)	0.60
Urine antigen ^4^	37/37 (100%)	11/14 (79%)	0.003
Serum antigen ^4^	22/22 (100%)	6/9 (66%)	0.004

(+) = Positive test; pt.= patients. ^1^ Ascitic fluid, bone marrow, lymph node, maxillary sinus tissue; lymph node and sinus tissue yielded *H. capsulatum.*
^2^ Bone marrow, pericardial tissue. ^3^
*Histoplasma* antibodies by complement fixation and immunodiffusion (Michigan Department of Health and Human Services Laboratory). ^4^ Enzyme immunoassay (Mira-Vista Laboratories).

## Data Availability

The original contributions presented in this study are included in the article. Further inquiries can be directed to the corresponding author.

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
