# Peer review of "Characteristics and Outcomes of Hospitalized Patients with Histoplasmosis: Comparison of Immunocompromised and Non-Immunocompromised Adult Patients"

_jof, 2025, doi:10.3390/jof11090671_

Round 1

Reviewer 1 Report

Useful review of inpatient histoplasmosis from an endemic area. Additional details and specifics are necessary, especially for this cohort which is limited to inpatients.

  • Disseminated histo: use the term 'non-contiguous sites'
  • Since SOT recipients were the main immune compromised cohort, need more details as a separate paragraph in the 'results' section: time from SOT to histo, induction therapy, rejection within the 3 months prior to histo requiring augmented immune suppression, any donor-derived infection, and did histo lead to graft loss or dysfunction. 
  • Line 117: 'all non-immmunocompromised patients with disseminated histo had chronic illnesses:" are these patients really not immune compromised? solid tumors - recent chemo (outside of your 'methods' definition'?), diabetes - did the patient have DKA or hyperosmolar coma that increases risk of IFI, sarcoid/RA - were they on current immune suppression that did not meet prespecified criteria?
  • Did any patient have pleural involvement/empyema?
  • Miliary findings on CT chest?
  • A total of 17 patients were fungemic (table 5). Need additional details: how many had endocarditis? Any endovascular hardware (valves, electronic devices, vascular grafts), did any need valve surgery? What were the outcomes of those who were fungemic? How many had TEE? 
  • Line 212: 20 patients were admitted to the ICU. How many needed mechanical ventilation? SOFA score? Any ECMO usage? 
  • Was autopsy performed? What were the findings?
  • HLH is quite common with disseminated in-patient histoplasmosis. How many had confirmed HLH? What was the outcome in such patients?
  • Since this is an in-patient histo manuscript, include references such as those suggested below and discuss extrapulmonary syndromes (IE, CNS, HLH) as it is very relevant to hospitalized severe histo patients.
  • Discussion: Section is very long and repeats findings already stated in the results. Current discussion section should be significantly trimmed. Instead include and discuss suggested items above and markers of severity and management of specific syndromes.
  • Favor including references with brief discussion:
  • https://doi.org/10.1093/ofid/ofab360
  • https://doi.org/10.1097/md.0000000000010245
  • https://doi.org/10.1097/md.0000000000000034
  • https://doi.org/10.3389/fcimb.2022.847950

Reviewer 2 Report

The manuscript "Characteristics and Outcomes of Hospitalized Patients with Histoplasmosis: Comparison of Immunocompromised and Nonimmunocompromised Adult Patients" is an interesting and generally well-structured work that analyzes, through a retrospective comparison of immunosuppressed and nonimmunocompromised patients, the role of immunosuppression in patients with newly diagnosed histoplasmosis. It contains epidemiological aspects that may be useful to the medical community and other related fields; however, I have some comments.

Introduction

I suggest that the authors mention what histoplasmosis is, and that a species complex causes this mycosis.

The authors mention “The manifestations of infection with Histoplasma capsulatum vary widely, from mild respiratory infection to severe pulmonary or disseminated illness, depending on the balance between exposure to the organism in the environment and the immune status of the host [1]”, however, it would also be important to mention other factors, such as the virulence of the fungus and the amount of inoculum in the host.

Line 44: The authors mention “Early studies called attention to the importance of the immune response, typified by the reticuloendothelial system, in response to exposure to H. capsulatum.” However, I suggest including a more up-to-date reference, since the term “reticuloendothelial system” is no longer used; the term “mononuclear phagocytic system” is currently used.

Table 1 does not include the number of female patients.

I suggest increasing the font size used in Figure 1.

In the caption to Figure 2, I suggest the authors mention that the geographic distribution corresponds to the state of Michigan.

Discussion

The authors mention: “However, histopathology is not always readily attainable, and growth of Histoplasma from tissue or body fluids can take up to six weeks, with a further delay for confirmation from a reference laboratory that the organism is H. capsulatum. Despite these limitations, cultures, especially those obtained from blood samples that generally yield growth more quickly than other samples, are particularly useful for confirming disseminated disease in immunocompromised patients.” Therefore, I believe that molecular testing is important in these cases; furthermore, metagenomics is currently being implemented in the diagnosis of this mycosis. Therefore, I suggest discussing why molecular methods were not used in this study.

I suggest that the authors discuss how fungal factors, such as virulence or inoculum size, may be related to the severity or clinical presentation in the host. It would also be interesting to discuss the importance of identifying the different species that can cause histoplasmosis.

Author Response

plsee see the attached file with response to both reviewers 

Round 2

Reviewer 1 Report

Authors have reasonably responded to my prior comments. Current version of the manuscript can be accepted. 

As above.